# Acute Kidney Injury in SARS-CoV-2 Infection: Direct Effect of Virus on Kidney Proximal Tubule Cells

**DOI:** 10.3390/ijms21093275

**Published:** 2020-05-05

**Authors:** Manoocher Soleimani

**Affiliations:** Medicine and Research Services, VA Medical Center, Department of Medicine, University of New Mexico Health Sciences Center, Albuquerque, NM 87106, USA; MSoleimani@salud.unm.edu

**Keywords:** acute kidney injury, SARS-CoV-2, proximal tubule cells, ACE2, lysosomes

## 1. Introduction

Coronaviruses (CoVs), including Severe Acute Respiratory Syndrome (SARS), Middle East Respiratory Syndrome (MERS), and the novel coronavirus disease-2 (SARS-CoV-2) are a group of enveloped RNA viruses that cause a severe respiratory infection which is associated with a high mortality [1,2]. Amongst these viruses, SARS-CoV-2 is the most virulent and is spreading rapidly around the world. In addition to the lungs, SARS-CoV-2 can affect the heart, kidney, and gastrointestinal tract [3,4]. Presently, more than 2.0 million people have been infected worldwide with over 180,000 deaths.

All coronaviruses utilize common cellular pathways to gain entry into the host cells. Patients predominantly present with an infection of the respiratory system, where the virus binds to the angiotensin-converting enzyme-2 (ACE2) in both the upper and lower airway epithelia in order to enter the cell [1]. In addition, co-receptors/auxiliary proteins from protease families such as TMPRSS2 (Trans Membrane Serine Protease 2) may work in collaboration with ACE2 by priming the viral S (spike) protein and therefore facilitating the entry of SARS coronaviruses into target cells, including airway epithelia [1,2]. Following the binding to the receptor (ACE2) and the co-receptor (TMPRSS2 or other proteases), the virus utilizes the host endocytosis machinery to fuse with the membrane and internalize [1,5]. The endocytosed virus is transported to the early endosome, late endosome, and eventually lysosome, where it becomes activated and acquires the capability to start replication and infection [1,5,6].

### 1.1. Acute Kidney Injury in SARS-CoV-2 Infection: Impact on Mortality

There is a high incidence of acute kidney injury (AKI) in patients with SARS-CoV-2 infection. In a very recent study of critically ill patients with SARS-CoV-2 pneumonia in Wuhan, China, 29% of those admitted to the hospital developed acute kidney injury [3]. In a separate study, out of 163 critically ill patients who recovered from the SARS-CoV-2 infection, only one patient had developed AKI, whereas out of the 113 patients who died, 28 patients had developed AKI during the course of their hospitalization [4]. These results indicate a much higher mortality rate in critically ill SARS-CoV-2 patients with AKI vs. those without AKI [3,7] *. In a large cohort of 536 SARS patients from London in 2005, 36 patients exhibited elevated plasma creatinine levels during their clinical course [7]. Those patients with AKI were older and had higher systolic blood pressure than other SARS patients with no AKI [7]. The post-mortem kidney histology showed acute tubular necrosis, with the majority of the damage detected in the proximal tubule, and no evidence of glomerular injury [7]. Patients with SARS-CoV-2 develop acute kidney injury (AKI), with a significant number exhibiting proteinuria and a smaller fraction displaying hematuria [3,4].

### 1.2. Acute Kidney Injury in SARS-CoV-2 Infection: Etiologies and Pathogenesis

Is CoV-AKI the result of altered hemodynamics (Ischemic Reperfusion Injury), direct viral damage, or both? The cause of AKI in SARS-CoV-2 is multi-factorial. Both sepsis-related and unrelated pathways are likely contributing to kidney injury in patients with SARS-CoV-2 infection. While those associated with sepsis (or septic shock) could develop kidney injury as a consequence of their altered hemodynamic status, it seems that a portion of kidney injuries occurs independent of sepsis or sepsis-related pathways **. It is plausible that kidney damage in the latter group reflects a significant effect of the virus on the kidney tubules. This assumption is based on several findings. First, SARS coronaviruses, including SARS-CoV-2, are detected in the urine via PCR where viral fragments are identified, indicating that the virus has a direct interaction with or exposure to the kidney tubules [7,8] ***. Second, the tubular expression pattern of ACE2 (which functions as a receptor for the virus) is limited to the proximal tubule [9,10] and parallels the sites of injury in the kidneys of patients with the SARS-CoV infection [7]. Lastly, SARS-CoV shedding in the urine was detected between the second and third week of the viral infection and correlated with the onset of AKI [7,11]. These data raise the possibility that SARS-CoV-2 may directly damage the kidney tubules. Given the strong expression of ACE2 along the apical membrane of proximal tubule cells, it is highly plausible that SARS-CoV gains entry access to the proximal tubule cells by binding with ACE2.

Unlike airway epithelial cells, where viral entry is dependent on the presence of ACE2 working in tandem with the viral S protein priming serine protease TMPRSS2 [1], kidney proximal tubules cells express very low levels of TMPRSS2 [12,13]. However, kidney proximal tubule cells express abundant levels of related proteases, including cysteine protease cathepsin B/L, glutamyl aminopeptidase, and serine protease dipeptidyl peptidase 4 (DPP4) [14,15,16], all of which are potential viral S (spike) priming proteases [2,16] and might substitute for TMPRSS2, thereby facilitating the entry of SARS-CoV-2 into the proximal tubule cells by working with ACE2. Whereas cathepsins could be potential collaborators for ACE2 in facilitating SARS-CoV’s entry [17], the role of DPP4 in SARS-CoV’s entry into cells has not been settled [1,18]. DPP4, however, may play a critical role in facilitating MERS-CoV entry into the pulmonary alveolar cells [19]. The kidney tropism of coronaviruses (both SARS-CoV and MERS-CoV) has been demonstrated by the robust virus replication in cultured kidney proximal tubule cells [20].

### 1.3. It Is Therefore Plausible That ACE2 and Specific Proteases (E.G., Cathepsins) Work Together to Facilitate the Entry of SARS-CoV-2 Tnto Kidney Proximal Tubule Cells

Following the binding to surface molecules (ACE2) on the plasma membrane of susceptible cells—such as macrophages, alveolar cells, or kidney proximal tubule cells—the SARS coronavirus internalizes into vesicles which traffic through the endosomal/lysosomal pathways [21,22]. Lysosomes are a key component of the endocytic pathway and contain a very high acidic luminal pH, which is generated by the actions of V-H^+^-ATPase, a proton-pumping transport protein, and the Na^+^/H^+^ exchanger isoform 6 (NHE-6) working in tandem with chloride channels (CLCs) [23]. The proposed schematic diagram in Figure 1 depicts the localization of ACE2 on the apical membrane and binding with the SARS-CoV-2 spike protein followed by the internalization of the virus protein and entry into lysosomes in the kidney proximal tubule cells.

In a detailed study examining the interaction of Ang-II (angiotensin II) and ACE2 in cultured (Neuro-2A) cells, the authors demonstrated that Ang-II treatment resulted in a significant attenuation of ACE2 enzymatic activity along with ACE2 internalization and degradation into lysosomes [24]. These effects were prevented by the lysosomal inhibitor leupeptin as well as the Ang-II type 1 receptor (AT_1_R) blocker losartan [24]. It is intriguing to examine the impact of leupeptin and Ang-II on the ACE2 expression and the enzymatic activity in cultured pulmonary alveolar and proximal tubule cells, where ACE2 is abundantly expressed on the apical membrane and may play an important role in the binding and internalization of SARS-CoV-2.

In addition, lysosomes contain many hydrolases and play an essential role in the maturation/degradation stage of autophagy, which plays a critical role in SARS-CoV-2 replication. The acidic pH-dependent endosomal proteases cleave the viral glycoprotein segments to cross the replication events [21,22]. Inhibiting endosomal acidification will interrupt the cleavage processes and prevent viral replication.

Three categories of pharmacologic agents which can inhibit endosomal/lysosomal acidification are available and have been used to prevent the intracellular replication of various viruses. The first class of agents is comprised of weak bases—such as chloroquine, ammonium chloride, and amantadine—which diffuse across endo/lysosome membranes (along their concentration gradient) and become protonated, thereby releasing the base equivalent OH^−^ and alkalinizing the endosomal/lysosomal internal environment. The second class of agents comprises inhibitors of vacuolar H^+^-ATPases (e.g., bafilomycin A1 and concanamycin A), which have been used successfully to prevent endosomal/lysosomal acidification and virus transport and replication. The last group is made up of the carboxylic ionophores, such as monensin, which exchange endocytic protons for cytoplasmic potassium and sodium, thereby alkalinizing the lysosomes.

### 1.4. Targeting the Endosomal/Lysosomal Acidification and Their Acidic pH-Dependent Proteases by the Known Therapeutic Agents Could Be an Effective Treatment in Preventing SARS-CoV-2 Replication and Cell Death in Kidney Proximal Tubule Cells

It is worth mentioning that patients with pre-existing kidney disease (chronic kidney disease (CKD)) will be at increased risk of developing AKI during SARS-CoV-2 infection due to a number of factors, including the possibility of volume depletion as a consequence of decreased fluid intake and increased loss of gastrointestinal fluid due to diarrhea, as well as the use of NSAIDs for headaches and myalgias. A recently published large prospective study demonstrated that the presence of kidney disease in patients with the SARS-CoV-2 infection on admission was associated with the development of AKI and increased mortality during hospitalization [25].

## 2. Conclusions

There is a high incidence of acute kidney injury (AKI) in patients with the severe acute respiratory syndrome coronavirus-2 (SARS-CoV-2) infection. The mortality rate of critically ill SARS-CoV-2 patients with AKI is three times higher than for those without AKI. Is coronavirus-AKI (CoV-AKI) the result of altered hemodynamics (Ischemic Reperfusion Injury), direct viral damage, or both? While altered hemodynamics is the presumed culprit in causing kidney failure in a SARS-CoV-2 infection, a significant portion of kidney injuries occurs independent of sepsis-related pathways. It is plausible that kidney damage in the latter group reflects specific effects of the virus on the kidney tubules. This assertion is based on the appearance of coronavirus particles in the urine, the tubular expression pattern of ACE2 (a receptor for the virus)—which parallels the sites of injury in the kidneys of patients with SARS-CoV infection—and the correlation of the timeline of SARS-CoV viral shedding in the urine with the onset of AKI (Kidney International, Feb. 2005). These data raise the strong possibility of direct damage to the kidney tubules by the SARS-CoV-2 virus. We propose that SARS-CoV-2 enters the proximal tubule cells by binding with ACE2, which works in collaboration with specific proteases (e.g., Cathepsins) that modify the spike (S) protein to facilitate the entry of the virus into cells across the apical membrane. Identifying the etiology and pathogenesis of kidney failure in SARS-CoV-2 infections can lead to specific therapies aimed at preventing the binding, internalization, or replication of the virus in proximal tubule cells. Ultimately, these therapies should modulate the severity of kidney failure and significantly improve clinical outcomes and mortality by alleviating the burden of AKI in critically ill SARS-CoV-2 patients.

While this manuscript was under review, a manuscript demonstrating the presence of SARS-CoV-2 particles in kidney proximal tubule cells in patients with SARS-CoV-2 infection and renal failure appeared online [26].

* There are numerous articles that are published daily on clinical presentations and laboratory data from patients with the SARS-CoV-2 infection. Although a few articles have indicated that AKI is not a major concern in SARS-CoV-2 patients, it should be noted that the data demonstrating the development of AKI and its association with enhanced mortality in critically ill SARS-CoV-2 patients is extremely convincing.

** It is also plausible that the damage by SARS-CoV-2 could make the proximal tubules cells more susceptible to the ischemic damage resulting from hemodynamic alterations.

*** Another issue is the frequency of the presence of the virus fragments in urine, with some studies detecting the particles in very few and some finding them in larger numbers of patients. This discrepancy may be due to a lack of standardized sample collection, the type of PCR technique (regular PCR vs. nested primer PCR vs. real time PCR), performing the test in patients with a low viral load or in the first 10 days of admission to the hospital, and in those without overt AKI. These discrepancies only highlight the necessity for the standardization of testing and the need for a clear understanding of the impact of this virus on the kidney structure and function.

## Figures and Tables

**Figure 1 ijms-21-03275-f001:**
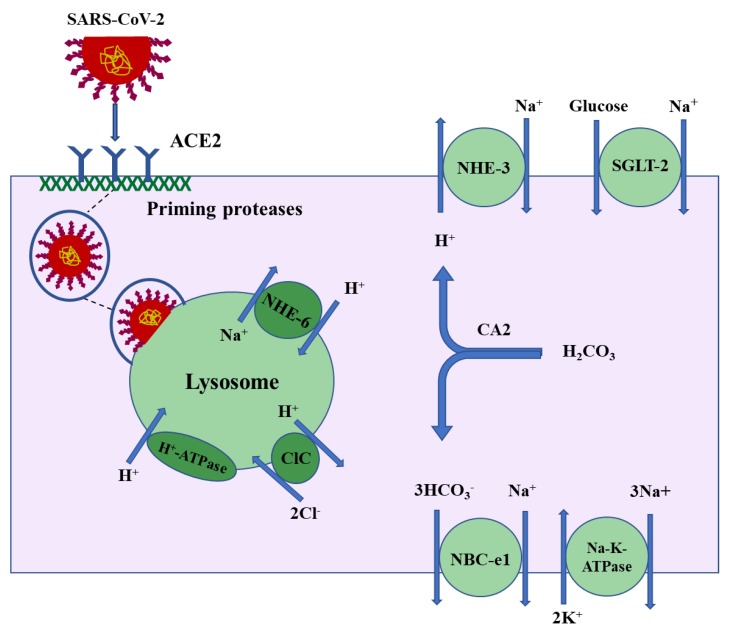
Proposed schematic diagram depicting the binding of SARS-CoV-2 spike protein to the apical membrane of the kidney proximal tubule cells followed by the internalization of the virus and its entry into lysosomes. The proximal tubule specific ion transporters are shown.

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
