# Peer review of "Acute Kidney Injury in SARS-CoV-2 Infection: Direct Effect of Virus on Kidney Proximal Tubule Cells"

_ijms, 2020, doi:10.3390/ijms21093275_

Round 1

Reviewer 1 Report

No doubt, COVID 19 is associated with the occurrence of AKI. However, COVID19-AKI is due to direct virus infection or it is a complication of other organ failure remains unclear. The evidence presented by the author was a bit weak. Publish this review after more original articles come out is believed able to significantly enhance the ms.  More data and references are needed.

Author Response

We thank the reviewer for the thoughtful critique. We agree with the referee that a review article on this topic is premature due to an extremely limited number of published studies. We also agree with the referee that the predominant belief in the field of nephrology is that the kidney failure in COVID-19 is largely due to multi-organ failure and altered hemodynamics. 

The current article was meant to be a Commentary or Perspective and the intention was to share with and alert the research community of the possibility that acute kidney injury in COVID-19 patients could be in part due to the direct damage by the virus on kidney tubules.

While this manuscript was under review, a manuscript demonstrating the presence of SARS-CoV-2 particles in kidney proximal tubule cells in patients with COVID-19 infection and renal failure appeared online in Kidney International (24). We hope that the referee agrees with the hypothesis that is advanced in our manuscript and finds the report in Kidney International (24) supporting that.   

If indeed the kidney failure is in part due to the direct effect of the virus, then identifying the pathways that would block the entry or replication of virus in kidney tubules could significantly alleviate the burden of kidney failure on mortality in critically ill COVID-19 patients. 

We have now added 12 new references in the bibliography section and discussed their relevance in the text. New additions are in red font. 

We hope that the referee finds the manuscript acceptable for publication.

Warm regards.

Reviewer 2 Report

There is evidence of the association between kidney involvement and poor outcome in patients with COVID-19, in particular patients with elevated baseline serum creatinine are more likely to be admitted to the intensive care unit and to undergo mechanical ventilation, suggesting that kidney disease on admission represents a higher risk of deterioration.

Could the authors please provide more details about the existing literature and the risk of death in patients with kidney injury, SARS and COVID19?

If this is going to be added, then the manuscript, already good, could be accepted pending minor revisions.

Author Response

We greatly appreciate the reviewer's thoughtful critique. While our paper was under review, a manuscript demonstrating the presence of virus particles in kidney proximal tubule cells along with renal failure in patients with COVID-19 infection appeared online in Kidney International (ref. 24; Pre-proof on April 9).

We have now discussed the role of CKD or baseline kidney disease on admission in patients with SARS-CoV-2 infection on the incidence of AKI and mortality during hospitalization (ref. 23).

We have included several references indicating increased incidence of AKI which was associated with a significant mortality in critically ill patients with SARS-CoV-2 or other coronaviruses  (3, 4, 7, 8, 23, 24).

In total, we have added 12 new references which encompass various areas ranging from increased incidence of AKI and mortality in hospitalized patients with COVID-19 infection, the increased risk of AKI and mortality in hospitalized COVID-19 patients with CKD or baseline renal disease on admission (ref. 23), the role ACE2 and DPP4 in binding and facilitating the internalization of SARS-CoV-2 in kidney proximal tubule,  and the role of lysosomal acidification on the activation and replication of coronaviruses. All new additions are in red font.

If indeed the kidney failure is in part due to the direct effect of the virus, then identifying the pathways that would block the entry or replication of virus in kidney tubules could significantly alleviate the burden of kidney failure on mortality in critically ill COVID-19 patients. 

We hope that the referee finds the manuscript acceptable for publication.

Warm regards.

Round 2

Reviewer 1 Report

The author addressed my concerns.